# Deep 3D Neural Network for Brain Structures Segmentation Using Self-Attention Modules in MRI Images

**DOI:** 10.3390/s22072559

**Published:** 2022-03-27

**Authors:** Camilo Laiton-Bonadiez, German Sanchez-Torres, John Branch-Bedoya

**Affiliations:** 1Facultad de Minas, Universidad Nacional de Colombia, Medellín 050041, Colombia; jwbranch@unal.edu.co; 2Facultad de Ingeniería, Universidad del Magdalena, Santa Marta 470001, Colombia; gsanchez@unimagdalena.edu.co

**Keywords:** medical image segmentation, deep learning, transformers, convolutional neural networks, brain structures

## Abstract

In recent years, the use of deep learning-based models for developing advanced healthcare systems has been growing due to the results they can achieve. However, the majority of the proposed deep learning-models largely use convolutional and pooling operations, causing a loss in valuable data and focusing on local information. In this paper, we propose a deep learning-based approach that uses global and local features which are of importance in the medical image segmentation process. In order to train the architecture, we used extracted three-dimensional (3D) blocks from the full magnetic resonance image resolution, which were sent through a set of successive convolutional neural network (CNN) layers free of pooling operations to extract local information. Later, we sent the resulting feature maps to successive layers of self-attention modules to obtain the global context, whose output was later dispatched to the decoder pipeline composed mostly of upsampling layers. The model was trained using the Mindboggle-101 dataset. The experimental results showed that the self-attention modules allow segmentation with a higher Mean Dice Score of 0.90 ± 0.036 compared with other UNet-based approaches. The average segmentation time was approximately 0.038 s per brain structure. The proposed model allows tackling the brain structure segmentation task properly. Exploiting the global context that the self-attention modules incorporate allows for more precise and faster segmentation. We segmented 37 brain structures and, to the best of our knowledge, it is the largest number of structures under a 3D approach using attention mechanisms.

## 1. Introduction

The scientific community has developed tools that allow for obtaining brain information so doctors can study the human brain. Among these tools we find brain imaging, which includes methods such as computed tomography (CT), magnetic resonance imaging (MRI), positron emission tomography (PET), and ultrasound (US), among others. However, not all methods produce quality images when applied to the brain because high contrast images are required to study the human brain. Therefore, highly sensitive methods such as magnetic resonance imaging or positron emission tomography must be used [1].

Although PET scans are capable of obtaining good quality images, they are not the preferred choice for specialists as they have several disadvantages. PET scans cannot reveal structural information at the microscopic and macroscopic levels in the white and gray matter of the brain; cannot detect changes in brain activation, and pose health risks due to the required radiation [2]. For these reasons, the MRI method has been widely used in the brain for medical studies and scientific research [3,4]. There are several types of MRI sequences that are capable of improving contrast and brightness in certain types of tissues. T1-weighted, T2-weighted, Fluid Attenuated Inversion Recovery (Flair), and Diffusion Weighted Imaging (DWI) are among the most common MRI sequences [5].

Traditionally, interpretations of medical images have been made by human experts. However, the existence of variations in the criteria among various human specialists is a limitation in relation to the generation of an efficient and precise diagnosis [6]. This weakness has been addressed through the development of computer aided diagnostic systems using computer vision for analysis of medical Images [7]. In this field, traditional algorithms were initially applied for the segmentation of brain anatomical structures, such as thresholding techniques [8], growth of regions [9], machine learning algorithms for classification [10], or grouping [11], among others. Based on the growth in computational capacity and the amount of data available, it is possible to use more robust and complex modern algorithms achieving better results in medical segmentation tasks [12].

In general, multiple methods for the segmentation of brain magnetic resonance images have been proposed. These methods can be grouped as follows: manual methods, spatial dimensionality-based methods (2D and 3D), pixel/voxel intensity-based methods, atlas-based methods, surface-based methods, and methods based on deep learning techniques [13].

The manual segmentation of magnetic resonance images is based on the use of highly trained personnel to obtain the different types of brain tissues. In order to perform segmentation, experts commonly use manual delineation tools that allow them to delineate different regions of the brain. Some examples of these tools are FreeSurfer [14], BrainSuite [15], FSL [16], ITK-SNAP [17], 3D Slicer [18], SPM [10], and Horos [19], among others.

In terms of the spatial dimensionality methods, these are subdivided into 3D and 2D approaches. Three-dimensional-based segmentation approaches seem to be the natural way to approach the problem because it allows to exploit the three-dimensional nature of MRI by considering each voxel and its relationship to neighbors at different acquisition planes (sagittal, coronal, and axial). However, the 3D approach still has limitations, mainly related to the high computational cost in computers with limited memory, the increase in the complexity of the models and the number of parameters, making the learning process slower [20,21,22]. Therefore, researchers usually use the 2D representation of a brain MRI to avoid memory restraints and computational limitations of the 3D representation.

Intensity-based methods attempt to find a threshold value that separates the different tissue categories. These methods include techniques of thresholding, growth of regions, classification, and grouping [8]. In [23,24], authors presented work on pixel intensity using thresholding techniques to segment brain tumors.

Works that use region growth techniques, use the similar characteristics of pixels found together to perform the separation of a common region [25]. Region growth techniques have been applied for the segmentation of brain tumors [9], organs [26], cerebral vessels [27], and lesions in both the brain and the breast, applying other techniques such as morphological filters [28] or with quantitative values such as the measurement of the roughness of the tumor border [29]. Within this group, we also find classification and grouping techniques that make use of labeled and unlabeled data for their operations. In fact, multiple brain segmentation tools that are widely used in the scientific community, such as FreeSurfer [14], 3D Slicer [18], and SPM [10] make use of Bayesian Classifiers. The k-means algorithm is the most used in clustering because it is simple to implement, relatively fast, and produces good results. Some examples of work for tissue or tumor segmentation are presented in [11] and in [30], where the k-means algorithm is combined with a vector support machine and a Fuzzy C-means algorithm with thresholding techniques, respectively.

Atlas-based methods are those that make use of brain anatomical information from a specific population for image segmentation. Likewise, in [31,32,33,34], it is shown that this method is not limited to the use of a specific atlas; multiple atlases and techniques such as tag fusion can also be used to improve the delimitation of brain regions.

Several works have also been proposed that make use of deformable models. These techniques are part of the group of surface-based methods where the main objective is the delimitation of regions with similar characteristics through the use of elastic curves [35]. Similar to the approaches mentioned above, surface-based methods have been used for the segmentation of brain regions [36] and tumors [37,38].

Recently, deep learning has become an area of interest in the scientific community due to the important results that have been achieved in multiple disciplines [39,40,41]. The strength of convolutional networks is that they automatically identify relevant characteristics without any explicit human supervision [42]. In addition, compared with their predecessors, fully connected neural networks, convolutional networks significantly reduce the number of trainable network parameters, facilitating computation and making it possible to build large-scale architectures. Likewise, they efficiently link the output of the network with the extracted characteristics by jointly training the classification layers and the characteristic extraction layers [43]. Specifically, in the problem of brain magnetic resonance imaging segmentation, deep learning has achieved better results than previously exposed methods [12]. Within the deep learning branch, there are multiple algorithms based on neural networks that have been developed with specific objectives, such as autoencoders [44], Boltzmann machines [45], recurrent neural networks [46], convolutional neural networks [47], and Transformers [48], among others. Convolutional neural networks are precisely the algorithms most widely used by researchers to perform image segmentation and classification tasks, given that they have achieved the best results to date.

Convolutional neural networks are a type of neural network that was created by LeCun but was inspired by Fukushima’s work on the neocognitron for the recognition of handwritten Japanese characters [49]. In the study of brain magnetic resonance imaging using neural network architectures, it is common to see convolutional neural networks as the basis of the architectures. In fact, in [50], the authors presented a solution for brain tissue segmentation on MRI images taken in infants approximately six to eight months of age using CNNs in the deep learning architecture.

Similarly, the authors of [51] were able to use CNNs for the segmentation of subcortical brain structures using the datasets of Internet Brain Segmentation Repository (IBSR) and LPBA40 [15].

Some of the most important deep learning solutions have been proposed using a 2D representation allowing researchers to segment more structures than a 3D representation allows them to do. In fact, the segmentation of more than 25 brain structures into a 3D representation has been achieved by a few works, while using a 2D representation, deep learning works can segment more than 95 brain structures [52].

The use of this type of neural network is not limited to the segmentation of brain tissues or brain structures. These have also been used in the segmentation of brain lesions, as in [53,54,55], the segmentation of brain tumors [56,57,58], the detection of ischemic strokes [59], and even genomic prediction of tumors [60]. The most important thing to note from these works is that many use branches within their neural network architectures. In general, they use two branches, where one of them is focused on the extraction of globally related characteristics (global context), while the other is in charge of the extraction of local characteristics (local context) to achieve better segmentation.

One of the architectures most commonly used in medical image segmentation tasks is the U-Net architecture [61]. Due to the structure of its architecture, U-Net has advantages over other convolutional neural network architectures of its time. This was built having a path that encodes the characteristics of the image and then continues with its expansion, that is, an encoder-decoder structure. In addition, to avoid the vanishing gradient and explosion problem, the U-Net architecture incorporates skip connections, between the encoder and decoder layers, which improves performance in small datasets compared with other architectures at the time.

Multiple neural network architectures based on U-Net have been proposed for the field of medical image segmentation. The primary goals of these works were to improve the network by using skip connections between the layers of the coding and expansion path [62,63] and to combine the architecture with others such as SegNet [64]. It is important to note that several of these studies were also applied to the segmentation of white matter, gray matter, and cerebrospinal fluid from brain magnetic resonance images.

However, convolutional neural networks have serious limitations. One of them is the loss of image characteristics due to pooling operations [65]. This is because the CNNs require these operations to reduce the feature maps resulting from the convolutions and thus reduce the computation required in subsequent layers. Due to this, a large amount of data is necessary in the training process for deep learning networks to be able to generalize and achieve good results [65].

On the other hand, researchers have proposed multiple deep learning architectures based on attention mechanisms such as the Transformer’s architecture [22,23,66]. This one was initially proposed in the field of natural language processing [67] as being in charge of transforming one sequence into another from multiple attention blocks [48]. The Transformers replaced the recurrent neural network models (RNN) used until then for the translation of texts because it solved its main weakness. This was because the performance of the recurring models fell when very long sequences were introduced due to the long-term dependency learning problem [66], and although this problem was attacked by the Long Short-Term Memory (LSTM) networks, they did not achieve as good results as the Transformers. This became possible since the latter, through self-attention mechanisms, are capable of processing the entire sequence entered, even if it is very long, optimizing processing times due to parallel processes within the network.

Thus, the scientific community has achieved that the Transformer’s architecture can obtain results comparable to those established as the state of the art in computer vision methods [68]. In fact, some methods based on Transformers’ architectures were proposed for the segmentation of medical images. The TransUNet [69] network, which is based on the U-Net architecture [61], consists of a set of convolutional layers to extract the most important characteristics of the image. The resulting feature maps are then the input to successive attention blocks, which then send this output to the decoder. The decoder is fabricated of convolutional and upsampling layers to achieve the output of a segmented image. It is necessary to mention that the set of convolutional layers is connected with the layers of the decoder through skip connections.

Also, another Transformer-based architecture is the Medical-Transformer network [70], which is based on the use of two branches for its operation. The important thing to highlight in this study is that it has a local and global branch, as has been proposed in various convolutional neural network architectures and the use of convolutions in the feature coding process. Specifically, the local branch has a greater number of encoder and decoder blocks than the global branch. The encoder blocks are fabricated of 1 × 1 convolutional layers, normalization, Rectified Linear Unit (ReLU) activations, and multiple layers of attention for its operation, while the decoder has closed axial attention layers.

In this study, a 3D architecture of deep neural networks is proposed for the task of segmenting volumes associated with brain structures from MRI. Our proposal uses an encoder/decoder approach, strengthening the connection between them by incorporating self-attention modules and skip connections. The attention modules as well as the convolution operations allow the network to incorporate global and local characteristics, and achieve a more detailed segmentation of the edges of structures.

## 2. Material and Methods

### 2.1. Dataset

We used the publicly available Mindboggle-101 dataset [71]. This dataset contains 101 manually labeled human brain images from healthy patients, containing more than 100 brain structures in each volumetric segmentation file aseg + aparc. In this dataset, brain structures are labeled based on the brain hemisphere where they are located (e.g., left hippocampus and right hippocampus).

### 2.2. Data Preprocesing

We performed some preprocessing steps. We created an array representation of the original MRIs and segmentation masks in the dataset making it straightforward for training deep learning models. Additionally, for the selected structures, we created a label remapping strategy with IDs 1 to 37 having the ID 0 set for background. We created the segmentation ground truth files by taking the aseg+aparc files and mapping the existing IDs to our desired IDs. After this, we applied a min-max scaling to the voxel values to be in a range between 0 and 1. This intensity rescaling was performed using the histogram equalization technique over each individual MRI.

Then, we applied a filter in each MRI where empty slides were removed from the brain volumes, leaving on average 192 slides per brain plane. These preprocessed MRI volumes were divided into nonoverlapping subvolumes of size 64 × 64 × 64 voxels that were saved in single files containing stacks of volumes per brain MRI (see Figure 1). Finally, we divided the dataset into two sets in the ratio of 8:2. The first set was for training the neural network architecture, and the second one was for validating it. Due to the Mindboggle-101 dataset contains multiple datasets such as OASIS-TRT-20 which contains 20 MRIs, NKI-22 that contains 22 MRIs, among others, we performed the dataset division maintaining the original dataset distribution, making sure that the validation set had at least one MRI from all the datasets.

### 2.3. Computational Resources

The computational implementations were performed with the open source library for numerical computation Tensorflow and run on a computer with a 5th generation Intel I7 4820k@3.70 GHz processor, 64 GB of RAM memory, and two Nvidia 1080TI video cards with 11 GB of GDDR 5x RAM at 405 MHz.

### 2.4. Model for Brain Structures Segmentation from MRI

The proposed deep neural network architecture is structured as an encoder-decoder architecture. The contracting path follows the typical architecture of a convolutional neural network. However, we applied Transformer layers at the end of this path using the extracted feature maps from the CNN layers. The expansive path was composed of a successive combination of convolutional neural networks and upsampling layers in order to reach the original spatial resolution (see Figure 2). To avoid gradient vanishing and explosion problems, we adopted skip connections between the encoder-decoder paths via the usage of Res paths, initially proposed in [64].

The architecture input is as follows: given an image 
x ∈ ℝH×W×D×C
 where *H* represents height, *W* width, *D* depth, and *C* the numbers of channels. The objective of this study is to segment multiple anatomical brain structures, predicting the corresponding pixel-wise label maps of size 
H×W×D
.

### 2.5. Self-Attention Mechanism for Brain Segmentation

We used self-attention mechanisms via Transformers in the encoder path. This consists of successive 
I
 layers of Transformers composed of Multi-Head Self-Attention (*MHSA*) modules and Multi-Layer Perceptron (MLP) blocks, each preceded by a normalization layer. The MLP blocks use the RELU activation function with a regularization dropout layer.

The attention mechanism was computed in parallel inside each of the heads of the *MHSA* modules in each transformer using a set of vectors named as query, key, and value vectors [48]. The query vector 
q ∈ ℝd
 was matched against all key vectors organized in a matrix 
K ∈ ℝk×d
 using an algebraic operation known as the dot product. The results were then scaled using a scaling factor 
$\frac{1}{\sqrt{d_{k}}}$
 and normalized using a *softmax* function to obtain the weights. The attention matrix inside each *MHSA* head is computed as:
(1)
$$Attention\left(Q,K,V\right)=softmax\left(\frac{QK^{T}}{\sqrt{d_{k}}}\right)V$$

where *Q*, *K*, and *V* are matrices representing a set of queries, keys, and values, respectively.

Finally, the results of each head were concatenated and linearly projected into a matrix sequence at the end of the *MHSA* module. This can be described as:
(2)
$$MHSA\left(Q,K,V\right)=Concat\left({\mathrm{head}}_{1},{\mathrm{head}}_{2},\ldots ,{\;\mathrm{head}}_{h}\right)W^{O}\;\mathrm{where}\;{\mathrm{head}}_{i}=Attention\left(QW_{i}^{Q},KW_{i}^{K},VW_{i}^{V}\right)$$

where the projections are parameter matrices 
WiQ ∈ ℝdmodel×dk, WiK ∈ ℝdmodel×dk, WiV ∈ ℝdmodel×dv
 and 
WO ∈ ℝhdv×dmodel
 is the dimension of the Transformer’s hidden layers.

In the proposed architecture, we used the extracted feature maps from previous convolutional layers as the input of the first transformer layer, using a trainable linear projection. Indeed, we reshaped the feature maps 
x∈ℝH×W×D×C
 into a flattened representation as the transformer layers expect a sequence as input. Then, we applied positional embedding over the feature maps to add location information for the segmentation process. This can be described as:
(3)
$$fx_{i}^{q,k,v}=FeatureMapsEmbedding\left(flatten\left(x_{i}\right)\right)$$

where positional embedding adds location information useful in the segmentation process. This can be seen as:
(4)
z0=[F1; F2; F3; ⋯; FN]+Epos,    F ∈ ℝ,  Epos ∈ ℝN×L

where 
F
 denotes the feature maps in conjunction with the linear projection and 
$E_{pos}$
 the position embedding and 
$N=\frac{H\times W\times D\times C}{16}$
. After successive layers of Transformers, the output of the last transformer has a shape 
zI∈ ℝd×N
. We applied a reshape before the decoder path to recover its 3D dimensionality.

### 2.6. Loss Functions and Class Weights

Segmentation of brain structures is a highly imbalanced problem due to the significant differences in size in the structures, presenting greater availability of information in the image for those of greater size. Even the size difference between the structures and the background is usually significant. Therefore, multiple loss functions and weighting strategies for loss functions were proposed for improving imbalanced brain structure segmentation [72]. In the proposed approach, we used a combination of Dice Loss [73] and Focal Loss [74].

Dice Loss (DL) has its origin in the Dice Similarity Coefficient (DSC), which is widely used as a metric for computer vision segmentation to calculate the similarity between two images. Later, in [73], it was adapted as a loss function useful in medical image segmentation tasks, improving the imbalance problem between foreground and background. It is formulated as:
(5)
$$\mathrm{Weighted}\;\mathrm{Dice}\;\mathrm{Loss}=1-2\frac{\sum_{j=1}^{S}w_{j}\sum_{i=1}^{N}y_{ij}p_{ij}}{\sum_{j=1}^{S}w_{j}\sum_{i=1}^{N}y_{ij}+p_{ij}}$$

where 
$w_{j}$
 is the weight of the 
jth
 brain structure and *S* is the total number of segmentation classes, 
$y_{ij}$
 is the label of voxel 
i
 to belong to brain structure 
j
 and 
$p_{ij}$
 is the probability of voxel 
i
 to belong to brain structure 
j
.

Meanwhile, Focal Loss (*FL*) is a variation in Binary Cross-Entropy that works better with highly imbalanced datasets. It down-weights the contribution of easy examples and mostly focuses on the hard ones. It can be described as follows:
(6)
$$FL\left(p_{t}\right)=-\alpha_{t}{\left(1-p_{t}\right)}^{\gamma }\log\left(p_{t}\right)$$

where 
$p_{t}$
 with 
p ∈[0, 1]
 is the model’s estimated probability for the class, 
${\left(1-p_{t}\right)}^{\gamma }$
 is the modulating term with 
γ
 as the focusing parameter that controls its strength.

In this study, we used a combination of these two functions.

### 2.7. Metrics

In order to evaluate the performance of the proposed segmentation method, the ground truth and model prediction from the MRIs were compared. The selected metric for this comparison is the *DSC* which can be seen as a harmonic mean of precision and recall metrics. This metric can be mathematically expressed as:
(7)
$$DSC\left(x,y\right)=2\times \frac{x\cap y}{x+y}$$

where 
x
 represents the image ground truth and 
y
 represents the predicted output for that image. The marked labels for 
x
 and 
y
 should be binary represented for each class where all voxels included in a given class should have a value of 1 and 0 for all the others. Therefore, the *DSC* must be calculated individually for each class having output values in a range between 0 (no segmentation) and 1 (perfect segmentation).

Consequently, *precision* is the accuracy of positive predictions while *recall* is the ratio of positive elements that were predicted correctly by the segmentation model. These metrics can be expressed as:
(8)
$$precision=\frac{TP}{TP+FP}$$


(9)
$$recall=\frac{TP}{TP+FN}$$

where 
TP
 is the number of true positive predictions, *FP* the number of false positive predictions and 
FN
 the number of false negative predictions for a given class.

Also, the Intersection over Union (*IoU*) metric is included in this study as an evaluation metric for specific structures. It is useful for comparing similarities between two shapes 
A
 and 
B
 and determining true positives and false positives from a set of predictions. It can be expressed as:
(10)
$$IoU=\frac{A\cap B}{A\cup B}$$


### 2.8. Training Process

The training process used a lineal learning rate schedule, initially set at 0.001 and decreased after the 12th iteration to a power of 0.5, while the batch size is set by default at 8. It used the Adam algorithm as the neural network optimizer. For the transformer architecture based on the Visual Transformer (ViT) architecture [75], we set the successive layers and heads per layer at 4, the hidden size at 64, the MLP size at 192, the dropout rate at 0.1, the normalization rate at 0.0001, and a patch resolution of 
8×8×8
. It is important to mention that the hyper-parameters were chosen via experimental design.

## 3. Results

### 3.1. Performance of Proposed Deep Neural Network Architecture

We quantitatively and visually evaluated the performance of brain structures segmentation.

Figure 3 shows the number of voxels for each of the 37 selected structures from Mindboggle-101 dataset. It can be seen that there are significant differences between classes. To mitigate the effect of class imbalance we use a loss function combining the weighted coefficient Dice and the Focal Loss.

The combination of these two loss functions helped us to alleviate the imbalance problem in the segmentation of anatomical brain structures and encourages the correct segmentation of tissue boundaries. Indeed, the use of class weights while training deep neural network architecture was necessary due to the large number of small structures the brain has compared with the total number of brain voxels. In order to calculate the class weights, we used the median frequency balancing algorithm, which is formulated as follows:
(11)
$$\alpha_{c}=median\_freq/freq\left(c\right)$$

where 
freq(c)
 is the total number of voxels of class c divided by the total number of voxels of the MRIs where class c appeared and 
median_freq
 is the median of the calculated frequencies.

A graphic example of the segmentation result of the proposed deep neural network architecture can be seen in Figure 4, where we show the results in the axial, sagittal and coronal planes. The local details at the edges of the structures as well as the global features can be noted compared with the reference image. Quantitatively, we calculated the Precision, Recall, and Dice Score per segmented brain structure (see Table 1). These results show that there are still problems with the segmentation of some structures, mainly small structures that tend to lower values of quality metrics.

Although to achieve the training process of the deep neural network architecture the brain is divided into blocks of equal size, the results show that the segmented structures maintain spatial coherence and can recover its representative organic form as can be seen in a 3D visual representation shown in Figure 5.

### 3.2. Patch Resolution Size Determination

For this experiment, we set four successive layers of transformers with patch resolution blocks of 
8×8×8
, hidden size at 64, MLP size at 192; dropout rate at 0.1, and normalization rate at 0.0001. Afterwards, we applied a reshape before the decoder path to recover its 3D dimensionality. Therefore, in order to obtain finer details (local information), it was necessary to use convolutional layers at the beginning of the encoder path following a classical U-Net [61]-based architecture with the proposed architecture.

We experimented with patch resolution sizes related to the transformer layers. As was initially observed in [69], patch resolution size is important since it dictates the number of complex dependencies that each element will have, with others, obtaining finer details in the segmentation process. The ideal case would be to have a patch resolution size of 
1×1×1
. Nevertheless, there are not enough computational resources to train a deep neural network architecture based on this patch resolution size. Consequently, we ran experiments on the segmentation of three structures with patch sizes of 
16×16×16
 and 
8×8×8
 to see its influence on the segmentation of brain structures (see Figure 6).

### 3.3. Comparison with Other Methods

At present, the majority of the proposed deep neural network architectures for brain segmentation using Transformers are oriented toward the segmentation of brain tumors. Therefore, it is highly difficult to have a fair comparison since these models were oriented to the segmentation of one class, and not for multiple classes excluding background. Because of this, we implemented the 3D U-Net architecture, using it as our baseline, with an identical experimental setup. This comparison using the Dice score and the Wilcoxon signed-rank test can be seen in Table 2.

The time needed to perform the segmentation by this architecture and the comparison with other deep learning models is shown in Table 3. It is important to mention that transformer layers, thanks to the self-attention mechanism, are capable of processing entire sequences in parallel, optimizing processing times. Unlike CPU processing units, the GPU architecture was specifically designed to process data in parallel, allowing the proposed model to take full advantage of computational resources and the Transformer’s processing pipeline.

## 4. Discussion and Future Work

This study presents a deep learning-based model for the segmentation of 37 brain structures using transformer models. This network was trained with the manually annotated dataset Mingboggle-101, which contains 101 MRIs with its respective segmentation files processed using the Desikan-Killiany-Tourville (DKT) protocol [71]. In the scientific community, it is common to find multiple approaches to perform the segmentation of MRIs. Therefore, this architecture was indirectly trained to perform segmentation based on the DKT protocol due to the used dataset.

Our architecture includes self-attention modules to strengthen the connection between the encoding and decoding phases based on convolutional neural networks. The capabilities of self-attention modules add to the model the possibility of retaining features across voxels in the input patches of the model. Unlike 2D-based models, the 3D architecture can find voxel relationships in the three different planes, maximizing the use of the spatial nature implicit in MRI.

In addition, the results of the proposed segmentation model show that the quality metrics have a wide range of values. For the Dice Score, for example, the values vary from 0.32 to values of 0.93, showing low-quality segmentations for some structures. We find that the lower values tend to be related to structures with smaller volumes. The geometry at the edges of the structures is a factor that we consider influences the quality of the segmentation. Structures with borders of highly variable geometry tend to have segmentations with more error. Simpler edge structures generally result in more stable quality segmentations. Our intuition in this regard is that this is due, mainly, to the class imbalance problem and lack of enough data to train the model for those structures specifically. For this reason, it is important in the future to explore other methods that allow addressing this problem, for example, improving the calculation of the weights of the classes used in the loss functions similar to what is performed in [72], or using additional data augmentation techniques to increase the samples of classes with less information. Another factor that we considered in the analysis is the fact that deep learning methods based on transformers lack the inductive biases inherent in CNNs requiring large amounts of data to be able to generalize well [75], so their usage in small-size medical datasets remains difficult without any internal modification in their self-attention module. Incorporating these modifications can allow us to improve the segmentation of a large number of highly unbalanced brain structures using a 3D approach.

The patch resolution size is a determining factor to obtain finer details at the edges of segmentations. The experiments show an inverse relationship with size, that is, the smaller the patch size, the more detailed segmentation is obtained at the edges; the larger the patch size, the segmentations tend to be less detailed. It must be considered not all structures have geometrically complex edges, there are structures with simpler geometrics. Therefore, a trade-off between the computational cost of reducing the patch size and the more detailed segmentation requirements must be considered. Given the 3D representation used in this study and the memory requirements, it was not possible to explore values smaller than 8 × 8 × 8.

The results show that our method uses less time for segmentation with a Mean Dice Score similar to those found in the state of the art. Additionally, the segmentation of more than 25 brain structures into a 3D representation is a difficult task that has only been reported by a few groups of authors [52] due to computational and memory limitations. However, it is not competitive in terms of the number of segments where the latest 2D deep learning-based approaches are able to segment more than 100 structures. Consequently, further study should be carried out to optimize the use of memory and computational resources in the proposed architecture to segment more brain structures with a strong focus on the transformer architecture.

Our method still has deficiencies related to the variation in the segmentation quality for different structures. Class imbalance, as well as the broad geometric nature of the edges are factors for which our method is still sensitive. The number of segmented structures is also a limitation, it is desirable to be able to segment a greater number of structures, especially compared with 2D-based approaches.

In future work, we will explore the existing computational and memory limitations in our proposed architecture with a high focus on the transformer layers to see whether a different tokenization of the patched feature maps can improve its performance and segment more brain structures.

## Figures and Tables

**Figure 1 sensors-22-02559-f001:**
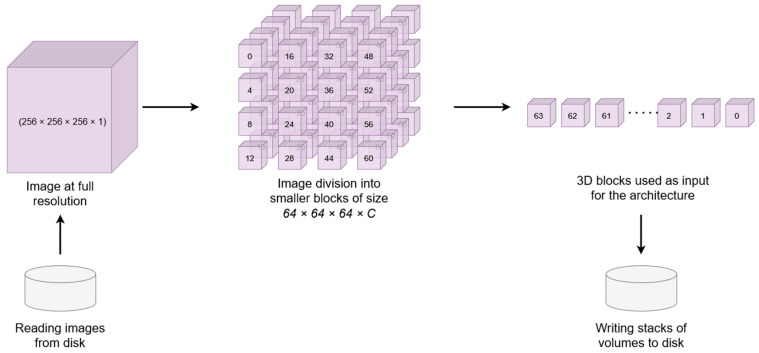
MRI volume divided into nonoverlapping subvolumes of size 64 × 64 × 64 voxels.

**Figure 2 sensors-22-02559-f002:**
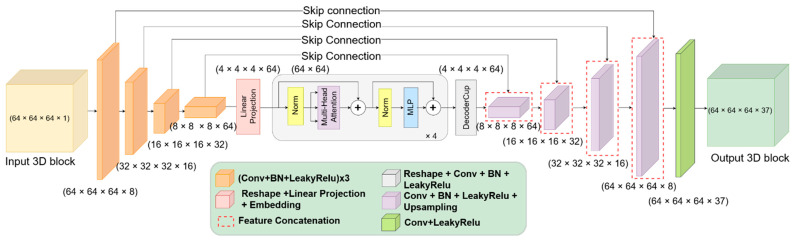
Deep neural network architecture for 3D brain MRI segmentation. The encoder part is composed of convolutional layers and Transformer layers, while the decoder has combinations of convolutional neural networks with upsampling layers.

**Figure 3 sensors-22-02559-f003:**
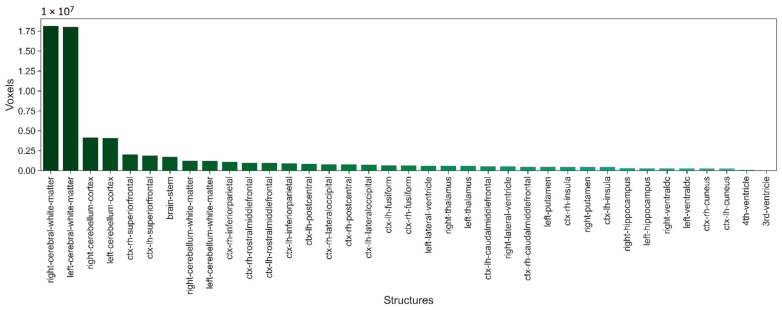
Number of voxels for each of the 37 selected structures from Mindboggle-101 dataset.

**Figure 4 sensors-22-02559-f004:**
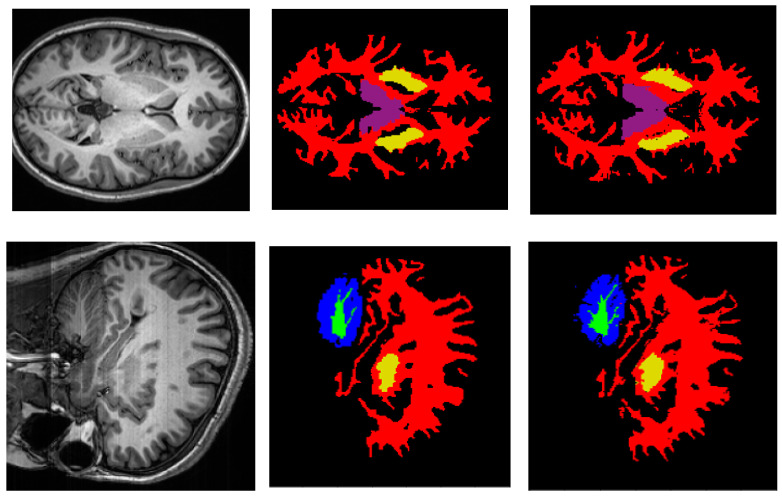
Segmentation results of the proposed architecture in the axial, sagittal and coronal planes where red, green, blue, purple, and yellow colors represent cerebral white matter, cerebellum white matter, cerebellum cortex, thalamus, and putamen structures, respectively. (**a**) Original MRI slide; (**b**) ground truth mask of the slide; (**c**) predicted MRI mask of the slide.

**Figure 5 sensors-22-02559-f005:**
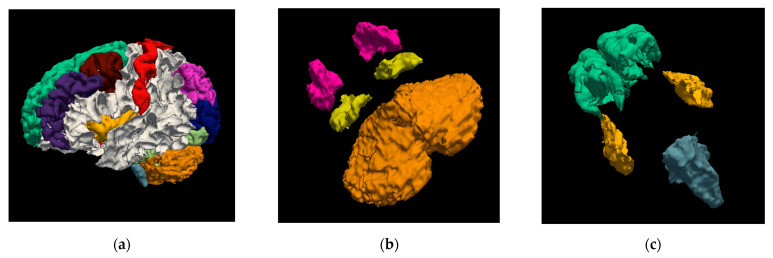
Segmentation results of the proposed architecture in 3D. (**a**) Segmentation of the 37 brain structures; (**b**) segmentation of the cerebellum cortex (orange), putamen (magenta), and hippocampus structures(yellow); (**c**) segmentation of the brain stem (gray), insula (yellow), and superior frontal structures (green).

**Figure 6 sensors-22-02559-f006:**
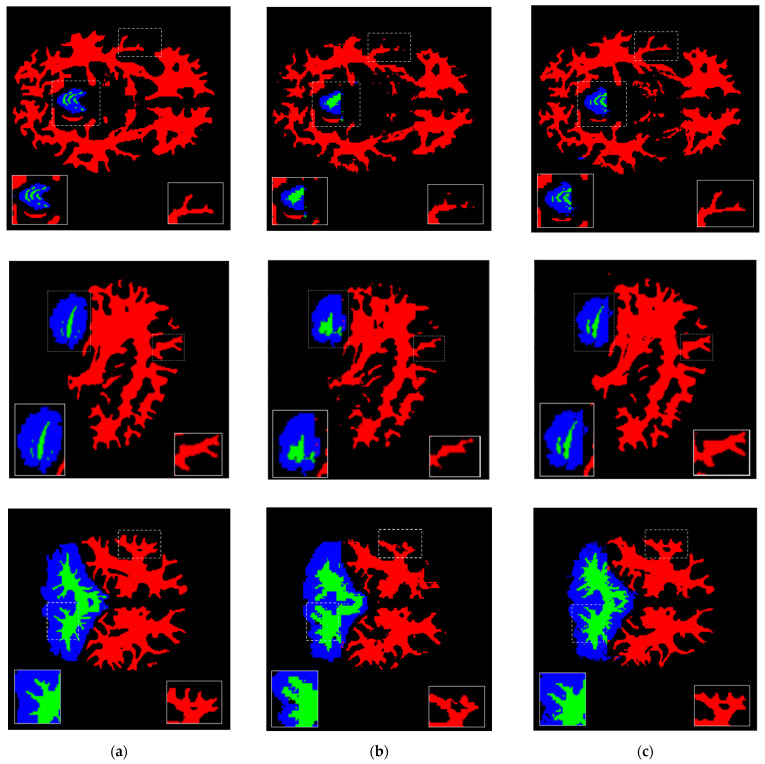
Patch size influence comparison on structure details segmentation in the axial, sagittal and coronal planes where red, green, and blue colors represent cerebral white matter, cerebellum white matter and cerebellum cortex structures, respectively. (**a**) Ground truth mask; (**b**) segmentation with patch resolution size of 
16×16×16
; (**c**) segmentation with patch resolution size of 
8×8×8
.

**Table 1 sensors-22-02559-t001:** Segmentation results per brain structure in a testing MRI.

Brain Structure	Precision	Recall	Dice Score	IoU
Left cerebral white matter	0.95	0.91	0.93	0.86
Right cerebral white matter	0.97	0.89	0.93	0.86
Left cerebellum white matter	0.90	0.75	0.82	0.69
Right cerebellum white matter	0.93	0.77	0.85	0.73
Left cerebellum cortex	0.87	0.82	0.84	0.73
Right cerebellum cortex	0.89	0.72	0.80	0.66
Left lateral ventricle	0.64	0.91	0.75	0.60
Right lateral ventricle	0.78	0.91	0.84	0.72
Left thalamus	0.80	0.92	0.86	0.74
Right thalamus	0.90	0.89	0.89	0.80
Left putamen	0.85	0.84	0.85	0.73
Right putamen	0.91	0.81	0.86	0.75
3rd ventricle	0.57	0.96	0.72	0.56
4th ventricle	0.67	0.94	0.78	0.64
Brain stem	0.87	0.93	0.90	0.83
Left hippocampus	0.88	0.67	0.76	0.62
Right hippocampus	0.89	0.80	0.84	0.73
Left ventral DC	0.78	0.83	0.80	0.68
Right ventral DC	0.62	0.87	0.72	0.57
Ctx left caudal middle frontal	0.84	0.43	0.57	0.40
Ctx right caudal middle frontal	0.50	0.24	0.32	0.20
Ctx left cuneus	0.56	0.65	0.60	0.44
Ctx right cuneus	0.54	0.74	0.62	0.46
Ctx left fusiform	0.68	0.61	0.64	0.48
Ctx right fusiform	0.78	0.65	0.71	0.55
Ctx left inferior parietal	0.64	0.54	0.58	0.42
Ctx right inferior parietal	0.60	0.70	0.65	0.49
Ctx left lateral occipital	0.69	0.74	0.71	0.56
Ctx right lateral occipital	0.73	0.69	0.71	0.56
Ctx left post central	0.54	0.82	0.66	0.49
Ctx right post central	0.71	0.70	0.71	0.55
Ctx right rostral middle frontal	0.57	0.81	0.67	0.50
Ctx left rostral middle frontal	0.51	0.82	0.63	0.46
Ctx left superior frontal	0.74	0.81	0.77	0.63
Ctx right superior frontal	0.77	0.79	0.78	0.65
Ctx left insula	0.81	0.84	0.82	0.70
Ctx right insula	0.70	0.87	0.78	0.64
Macro average	0.75	0.78	0.75	0.63
Weighted average	0.97	0.97	0.97	0.95

**Table 2 sensors-22-02559-t002:** Comparison between methods by the Dice Score and *p*-value for the Wilcoxon signed-rank test comparing proposed-UNet, proposed-DenseUNet samples pairs using the Mindboggle-101 dataset.

Model	Brain Structures	Mean Dice Score	*p*-Value
UNet (baseline)	37	0.790 ± 0.0210	0.0012850
DenseUNet (finetuned)	102	0.819 ± 0.0110	0.0211314
Proposed model	37	0.900 ± 0.0360	-

**Table 3 sensors-22-02559-t003:** Segmentation time per brain structure for a single MRI scan.

Model	Segments	Time Per Brain Structure	Mean Dice Score
DeepNAT [76]	27	~133 s (on a Multi-GPU Machine)	0.906
QuickNAT [77]	27	~0.74 s (on a Multi-GPU Machine)	0.901
DenseUNet	102	0.64 s (±0.0091 s) (Single GPU Machine)	0.819
FreeSurfer [77]	~190	~75.8 s	-
Proposed model	37	0.038 s (±0.0016 s) (on a Multi-GPU Machine)	0.903

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
