# Peer review of "Deep 3D Neural Network for Brain Structures Segmentation Using Self-Attention Modules in MRI Images"

_sensors, 2022, doi:10.3390/s22072559_

Round 1
Reviewer 1 Report
In this study, the authors proposed a deep learning architecture for brain structures segmentation. The idea is from self-attention modules and the authors modified the architecture to be suitable to this problem. Although the idea looks interesting, some major concerns should be addressed:
1. There must have some external validation data on the model.
2. Uncertainties of model should be reported.
3. When comparing the performance among methods, the authors should conduct some statistical tests to see significant differences.
4. More discussions should be added. The current discussion is not a formal discussion and lacks a lot of information.
5. Table 3 should include the comparison in terms of measurement metrics also.
6. Besides current metrics, the authors should report IoU also.
7. Deep neural network has been used in previous studies i.e., PMID: 34915158, PMID: 34812044. Thus, the authors are suggested to refer to more works in this description to attract a broader readership.
8. Abbreviations should be defined at the first use (i.e., convolutional neural network, etc.).
9. Source codes should be provided for replicating the study.
10. English language should be minor checked and fixed.
Author Response
We appreciate the comments of the reviewer. We believe that all of them have been completely useful to improve the quality of the paper, therefore all of them were addressed with changes within the paper. We have attended each of them, obtaining a completely restructured version of the article. All sections have been revised and modified to avoid repetitive ideas and deficiencies in the arguments.
We provide a description of the changes in the attached file

Reviewer 2 Report
The authors present a 3D segmentation strategy based on deep learning for brain structures segmentation in MRI images. The paper is clear, well written and results are sound. The manuscript adequately describes the background, present status and significance of the study. The authors compare their method with previously published papers. I have only one minor comment: please add the input size of the CNN in Figure 1.
Author Response

(The authors gave the same response as above.)

Reviewer 3 Report
The manuscript submitted by Laiton-Bonadiez and co-authors describes a segmentation technique for volumetric datasets of human brains. The technique is based on a combination of techniques that provide interesting results. These techniques are common and used, but the combination is not trivial and thus it is an interesting proposal. Whilst the results are encouraging, the way the manuscript is written is a bit repetitive and would need to be substantially re-written before it can be published. Specifically, manuscripts should follow the structure Abstract, Introduction, Materials and Methods, Results, Discussion. At the moment the manuscript is rather mixed, repetitive and deficient in some parts:
The abstract should include the results obtained in the work. These are not present.
Introduction and Background overlap. These two sections should be merged into one. The current introduction is lacking many references, which may be complemented with those in the background. The introduction should finish with a brief description of what is proposed, and the logic behind it. The phrase “Because of the aforementioned arguments …” is the correct start, but it falls short of a description. Some claims are not well justified, e.g. “3D structural brain segmentation is an approach that is still a problem in the scientific 71 community due to the existing computational limitations and memory constraints.” A quick google search showed more than 100,000 entries under volumetric segmentation brain MRI https://scholar.google.co.uk/scholar?hl=en&as_sdt=0%2C5&q=volumetric+segmentation+brain+MRI&btnG=
And similarly, there are entries on other areas of MRI analysis in volumetric perspective
https://scholar.google.co.uk/scholar?hl=en&as_sdt=0%2C5&q=volumetric+texture+MRI
https://scholar.google.co.uk/scholar?hl=en&as_sdt=0%2C5&q=volumetric+mri+analysis&oq=volumetric+MRI+brain
Introduction should be followed by materials (first) and methods (second). At the moment we find the materials inside the section “results”, which is not correct. Results are results, materials are materials. The methods should follow with a factual description of the methods applied in the paper with the spirit that these can be reproduced by the scientific community. If the code could be shared via GitHub or other platforms it would be better.
Figure 1 is good, albeit the fonts become a bit small, perhaps expand into two rows. On the other hand, figure 2 is rather vague, what are the readers supposed to learn from this?
Table 1 as it is, is rather useless. Unless someone is extremely familiar with that particular dataset it is not clear if 0.95 is good or bad, neither why the cuneus gets lower values than the insula. For this table to be useful, the authors could show what values are obtained by their work as compared with other works. Then it would be useful to show to indicate that the solution is good (or not) and why some areas are more difficult to segment should also be discussed.
As a reader Fig 3 is interesting, but I am clueless of what the colours represent. If the colours are important, these should be described in the caption. If they are not important, then a single colour could be used. In addition to the GT and prediction, a fourth column should be added, with a pixel-to-pixel comparison of the results, to show graphically the True Positives, True Negatives, False Positives / False negatives. Without this it is very very difficult, even for experienced researchers to evaluate how good or bad the results are. Similar case with Fig 4, what am I to perceive here? The authors should guide the reader “it should be noted that …”
Author Response
We appreciate the comments of the reviewer. We believe that all of them have been completely useful to improve the quality of the paper, therefore all of them were addressed with changes within the paper. We have attended each of them, obtaining a completely restructured version of the paper. All sections have been revised and modified to avoid repetitive ideas and deficiencies in the arguments.
We provide a description of the changes in the attached file

Round 2
Reviewer 1 Report
My previous comments have been addressed well.
Author Response
Thank you very much for your contributions. We are truly grateful for helping to improve the quality of the paper.
Reviewer 3 Report
sensors-1608886-peer-review-v1
The manuscript submitted by Laiton-Bonadiez and co-authors describes a segmentation technique for volumetric datasets of human brains. The technique is based on a combination of techniques that provide interesting results. These techniques are common and used, but the combination is not trivial and thus it is an interesting proposal. Whilst the results are encouraging, the way the manuscript is written is a bit repetitive and would need to be substantially re-written before it can be published. Specifically, manuscripts should follow the structure Abstract, Introduction, Materials and Methods, Results, Discussion. At the moment the manuscript is rather mixed, repetitive and deficient in some parts:
The abstract should include the results obtained in the work. These are not present.
Introduction and Background overlap. These two sections should be merged into one. The current introduction is lacking many references, which may be complemented with those in the background. The introduction should finish with a brief description of what is proposed, and the logic behind it. The phrase “Because of the aforementioned arguments …” is the correct start, but it falls short of a description. Some claims are not well justified, e.g. “3D structural brain segmentation is an approach that is still a problem in the scientific community due to the existing computational limitations and memory constraints.” A quick google search showed more than 100,000 entries under volumetric segmentation brain MRI https://scholar.google.co.uk/scholar?hl=en&as_sdt=0%2C5&q=volumetric+segmentation+brain+MRI&btnG=
And similarly, there are entries on other areas of MRI analysis in volumetric perspective
https://scholar.google.co.uk/scholar?hl=en&as_sdt=0%2C5&q=volumetric+texture+MRI
https://scholar.google.co.uk/scholar?hl=en&as_sdt=0%2C5&q=volumetric+mri+analysis&oq=volumetric+MRI+brain
Introduction should be followed by materials (first) and methods (second). At the moment, we find the materials inside the section “results”, which is not correct. Results are results, materials are materials. The methods should follow with a factual description of the methods applied in the paper with the spirit that these can be reproduced by the scientific community. If the code could be shared via GitHub or other platforms it would be better.
Figure 1 is good, albeit the fonts become a bit small, perhaps expand into two rows. On the other hand, figure 2 is rather vague, what are the readers supposed to learn from this?
Table 1 as it is, is rather useless. Unless someone is extremely familiar with that particular dataset it is not clear if 0.95 is good or bad, neither why the cuneus gets lower values than the insula. For this table to be useful, the authors could show what values are obtained by their work as compared with other works. Then it would be useful to show to indicate that the solution is good (or not) and why some areas are more difficult to segment should also be discussed.
As a reader, Fig 3 is interesting, but I am clueless of what the colours represent. If the colours are important, these should be described in the caption. If they are not important, then a single colour could be used. In addition to the GT and prediction, a fourth column should be added, with a pixel-to-pixel comparison of the results, to show graphically the True Positives, True Negatives, False Positives / False negatives. Without this it is very very difficult, even for experienced researchers to evaluate how good or bad the results are. Similar case with Fig 4, what am I to perceive here? The authors should guide the reader “it should be noted that …”
Author Response

(The authors gave the same response as above.)
